# Novel Thiazole Phenoxypyridine Derivatives Protect Maize from Residual Pesticide Injury Caused by PPO-Inhibitor Fomesafen

**DOI:** 10.3390/biom9100514

**Published:** 2019-09-20

**Authors:** Li-Xia Zhao, Min-Lei Yin, Qing-Rui Wang, Yue-Li Zou, Tao Ren, Shuang Gao, Ying Fu, Fei Ye

**Affiliations:** Department of Applied Chemistry, College of Science, Northeast Agricultural University, Harbin 150030, China; zhaolixia@neau.edu.cn (L.-X.Z.); minlei@aliyun.com (M.-L.Y.); qingruiwang@hotmail.com (Q.-R.W.); zouyuelineau@hotmail.com (Y.-L.Z.); woshibanye@gmail.com (T.R.); gaoshuang@neau.edu.cn (S.G.)

**Keywords:** PPO, thiazole phenoxypyridines, synthesis, safener activity, molecular docking

## Abstract

The herbicide fomesafen has the advantages of low toxicity and high selectivity, and the target of this compound is protoporphyrinogen IX oxidase (PPO, EC 1.3.3.4). However, this herbicide has a long residual period and can have phytotoxic effects on succeeding crops. To protect maize from fomesafen, a series of thiazole phenoxypyridines were designed based on structure–activity relationships, active substructure combinations, and bioisosterism. Bioassays showed that thiazole phenoxypyridines could improve maize tolerance under fomesafen toxicity stress to varying degrees at a dose of 10 mg·kg^−1^. Compound **4i** exhibited the best effects. After being treated by compound **4i**, average recovery rates of growth index exceeded 72%, glutathione content markedly increased by 167% and glutathione S-transferase activity was almost 163% of fomesafen-treated group. More importantly, after being treated by compound **4i**, the activity of PPO, the main target enzyme of fomesafen, recovered to 93% of the control level. The molecular docking result exhibited that the compound **4i** could compete with fomesafen to bind with the herbicide target enzyme, which consequently attained the herbicide detoxification. The present work suggests that compound **4i** could be developed as a potential safener to protect maize from fomesafen.

## 1. Introduction

Protoporphyrinogen IX oxidase (PPO) is a peroxidase that is immobilized in the chloroplasts of plants and is the pivotal enzyme involved in the biosynthesizes of chlorophyll. The main function of PPO is to catalyze the oxidation of protoporphyrinogen IX to the highly conjugated protoporphyrin IX [1,2,3]. PPO inhibitors can inhibit the occurrence of this process, causing the substrate protoporphyrinogen IX to accumulate in the cytoplasm, which in turn leads to lipid peroxidation of side chains and eventually leads to weed death [4,5]. As a type of efficient and selective PPO-inhibiting herbicide, fomesafen is used mainly in soybean fields, rubber plantations, and fruit orchards to control broadleaf weeds [6]. Fomesafen has the advantages of high efficiency, low toxicity, and environmental safety. However, fomesafen residues can remain in the soil for long durations and have phytotoxic effects on succeeding crops such as maize, which may inhibit growth and reduce the yield of maize [7,8]. With the introduction of fomesafen in soil at different concentration, the plant height of maize was inhibited to 56.6% to 85.8% of the control [9].

To protect crops from herbicides, various strategies have been employed, including research on readily degradable herbicides and decrease use of long-acting herbicides [10]. In addition, the use of herbicide safeners is an effective method to prevent crops from being damaged by herbicides [11,12]. Safeners can increase crop tolerance to selectively decrease the damage caused by herbicides [13]. Many types of herbicides contain compatible commercial safeners that can alleviate the phytotoxic effects of the herbicides against crops. For example, benoxacor can protect maize from metolachlor, and isoxadifen-ethyl can protect rice from fenoxaprop-P-ethyl [14]. To the best of our knowledge, very few researchers have investigated safeners for PPO-targeting herbicides.

Studies have shown that many safeners share common molecular characteristics, such as furilazole and AD-67, which have oxazolidine structures. Moreover, some safeners are structurally similar to the herbicides that they act on and can protect plants by competing with herbicides for target sites [15]. Therefore, commercially available pesticides can be used as intermediates or starting compounds, and lead compounds can be designed according to scientific methods [16,17,18]. In addition, structure–activity relationships, active substructure combinations, and bioisosterism have already been applied to design novel herbicide safener [19,20,21,22].

To obtain a safener for fomesafen, the active structure of fomesafen was studied. The structure of the diphenyl ether group in fomesafen, shared by all diphenyl ether herbicides, was removed and replaced by phenoxypyridine via a bioisosteric strategy [23,24,25,26]. As a mature safener, furilazole exhibited good biological activity and had an oxazolidine reactive group in common with other safeners [27,28]. According to their bioelectronic properties, the isosteric oxazolidine and thiazolidine may have similar activities and have also been reported to exhibit safener activity [29]. Based on the above studies and research, to obtain an active safener against fomesafen, a series of thiazole phenoxypyridine compounds **4a–4z** were designed based on structure–activity relationships, active substructure combinations, and bioisosterism (Figure 1).

## 2. Materials and Methods

### 2.1. Materials and Instruments

Maize seeds (Dongnong 259) were provided by Northeast Agricultural University college of agriculture. Fomesafen was provided by Jiangsu Fengshan Group Co., Ltd., the active ingredient content of 15% (m/m). The tested soil was Mollisols-cryolls clay loam type and collected from the Northeast Agricultural University Horticulture Station with a pH of 7.37. All the chemical reagents (Energy Chemical, Aladdin, China) were commercially available and used without further purification. Solvents were dried using standard methods. Thin-layer silica gel chromatography (TLC) (Qingdao Haiyang Chemical Co., Ltd., Qingdao, China) was used for analysis. The ^1^H-NMR and ^13^C-NMR spectra were recorded on a Bruker AV-300 spectrometer (Bruker Inc., Beijing, China) in CDCl_3_, internal standard TMS. High-resolution mass spectrometry (HRMS) data were recorded on a FTICR-MS spectrometer (Bruker Inc., Beijing, China). Melting points were measured on a X-4 equipment (Beijing Taike Inc., Beijing, China) and were uncorrected. X-ray diffraction data were collected on a D8 VENTURE X-diffractometer (Bruker Inc., Beijing, China).

### 2.2. Synthetic Chemistry

#### 2.2.1. General Method for Preparation of 2-Phenoxynicotinic Acid **1**

*N,N*-dimethylformamide (80 mL), phenol (0.125 mol), anhydrous K_2_CO_3_ (0.125 mol), and 2-chloronicotinic acid (0.1 mol) were sequentially added to a 250 mL flask with stirring. After adding CuCl (0.5 g) as a catalyst, the suspension was reacted under reflux. The reaction was allowed to proceed for 2 h before being cooled to room temperature. Water (150 mL) was added with stirring, and the pH was adjusted to 2–4 with hydrochloric acid to yield solids. The solids were recrystallized from ethanol to remove excess phenol and obtain a white solid product [30] (Scheme 1, Step a).

#### 2.2.2. General Method for Preparation of 2-phenoxynicotinoyl Chloride **2**

2-phenoxynicotinic acid (30 mmol), anhydrous CH_2_Cl_2_ (60 mL), and thionyl chloride (90 mmol) were sequentially added to a stirred solvent of anhydrous CH_2_Cl_2_ (60 mL). Then, five drops of *N,N*-dimethylformamide were added as a catalyst, and the reaction was refluxed for 1 h. A small amount of the reaction mixture was removed and added to methanol for TLC detection to determine whether the reaction had reached completion. CH_2_Cl_2_ (30 mL) was added to remove thionyl chloride under reduced pressure (Scheme 1, Step b).

#### 2.2.3. General Method for Preparation of Thiazolidines **3**

K_2_CO_3_ (30 mmol), toluene (30 mL), and cysteine hydrochloride (3.43 g, 20 mmol) were added sequentially to a 150 mL flask under stirring at ambient temperature for 10 min. Then, an appropriate ketone (30 mmol) was added dropwise. The reaction was allowed to proceed at 90 °C for 1 h with nitrogen as protection and cooled before further processing. Appropriate amounts of ethyl acetate were added, and the sample was washed three times with saturated NaCl solution (3 × 25 mL). Anhydrous magnesium sulfate was used to dry the organic phase and was removed by vacuum filtration. Solvents were evaporated with rotary evaporation, and the mixtures were separated by column chromatography to yield methyl thiazolidine-4-carboxylate [31] (Scheme 2).

#### 2.2.4. General Method for Preparation of Thiazole Phenoxypyridine **4**

Thiazolidine (5 mmol), K_2_CO_3_ (6 mmol) and CH_2_Cl_2_ (20 mL) were added to a 250 mL flask, then 2-phenoxynicotinoyl chloride (5 mmol) was added slowly with stirring. After the reaction was allowed to proceed at 30 °C for 30 min, K_2_CO_3_ was removed by filtration. Then, the contents were washed with saturated aqueous sodium chloride solution (3 × 20 mL), dried over anhydrous magnesium sulfate, and filtered to obtain the organic phase. The solvent was removed under reduced pressure, and the residue was separated by flash chromatography (1:4 EtOAc-ether) to obtain compound **4** as a colorless solid (Scheme 3).

*Methyl (R)-2,2-dimethyl-3-(2-(3-(trifluoromethyl)phenoxy)nicotinoyl)thiazolidine-4-carboxylate**(***4a***)* White solid, m.p. 84–85 °C; IR (KBr, cm^−1^) *ν*: 3044–2828 (C-H), 1722, 1620 (C=O), 1562–1405 (C=C), 1269 (C-O); ^1^H NMR (400 MHz, CDCl_3_) δ 8.18 (dd, *J* = 4.8, 1.9 Hz, 1H, Py-H), 7.80 (dd, *J* = 7.4, 1.6 Hz, 1H, Py-H), 7.53 (dd, *J* = 14.2, 7.8 Hz, 2H, Ar-H), 7.44 (s, 1H, Ar-H), 7.36 (d, *J* = 7.8 Hz, 1H, Ar-H), 7.10 (dd, *J* = 7.4, 5.0 Hz, 1H, Py-H), 4.79 (dd, *J* = 5.4, 2.1 Hz, 1H, N-CH), 3.70 (s, 3H, O-CH_3_), 3.35 (m, 2H, S-CH_2_), 2.02 (d, *J* = 33.4 Hz, 6H, C-(CH_3_)_2_); ^13^C NMR (101 MHz, CDCl_3_) δ 170.41, 164.50, 153.48, 148.31, 138.69, 130.23, 124.69, 122.93, 121.90, 121.86, 119.40, 118.38, 118.34, 73.54, 67.01, 52.94, 31.43, 29.06, 27.90; HRMS calcd. for [M + H^+^] C_20_H_20_F_3_N_2_O_4_S: 441.1090, found 441.1095.

*Methyl (R)-4-(2-(3-(trifluoromethyl)phenoxy)nicotinoyl)-1-thia-4-azaspiro [4.5]decane-3-carboxylate**(***4b***)* White solid, m.p. 121–122 °C; IR (KBr, cm^−1^) *ν*: 2939 (C-H), 1735, 1638 (C=O), 1561–1405 (C=C), 1274 (C-O); ^1^H NMR (400 MHz, CDCl_3_) δ 8.22–8.13 (m, 1H, Py-H), 7.77 (s, 1H, Py-H), 7.52 (dd, *J* = 13.4, 7.4 Hz, 2H, Ar-H), 7.44 (s, 1H, Ar-H), 7.35 (d, *J* = 7.2 Hz, 1H, Ar-H), 7.10 (t, *J* = 5.2 Hz, 1H, Py-H), 4.82 (s, 1H, N-CH), 3.71 (s, 3H, O-CH_3_), 3.25–3.02 (m, 2H, S-CH_2_), 2.61–1.92 (m, 4H, C-CH_2_-), 1.66–1.51 (m, 4H,-C-CH_2_-), 0.95–0.99 (m, 6H, -C-CH_3_); ^13^C NMR (101 MHz, CDCl_3_) δ 170.56, 164.64, 153.53, 148.17, 130.20, 124.66, 121.85, 119.42, 118.32, 67.09, 52.91, 36.02, 34.43, 30.81, 25.48, 25.41, 24.54; HRMS calcd. for [M + H^+^] C_24_H_28_F_3_N_2_O_4_S: 497.1716, found 497.1716.

*(2,2-dimethylthiazolidin-3-yl)(2-(3-(trifluoromethyl)phenoxy)pyridin-3-yl)methanone**(***4c***)* White solid, m.p. 70–71 °C; IR (KBr, cm^−1^) *ν*: 2963 (C-H), 1620 (C=O), 1563–1413 (C=C), 1246 (C-O); ^1^H NMR (400 MHz, CDCl_3_) δ 8.20 (dd, *J* = 5.0, 1.9 Hz, 1H, Py-H), 7.78 (dd, *J* = 7.4, 1.9 Hz, 1H, Py-H), 7.58–7.42 (m, 3H, Ar-H), 7.37 (dt, *J* = 7.9, 1.8 Hz, 1H, Ar-H), 7.13 (dd, *J* = 7.3, 4.9 Hz, 1H, Py-H), 3.90 (s, 2H, N-CH_2_), 3.01 (t, *J* = 6.0 Hz, 2H, S-CH_2_), 1.99 (s, 6H, C(CH_3_)_2_); ^13^C NMR (101 MHz, CDCl_3_) δ 163.92, 157.71, 153.54, 148.10, 137.59, 130.15, 124.79, 123.59, 121.79, 121.75, 119.40, 72.31, 54.36, 28.46; HRMS calcd. for [M + H^+^] C_18_H_18_ F_3_N_2_O_2_S: 383.1036, found 383.1038.

*Methyl (R)-4-(2-(3-(trifluoromethyl)phenoxy)nicotinoyl)-1-thia-4-azaspiro [4.5]decane-3-carboxylate**(***4d***)* White solid, m.p. 127–128 °C; IR (KBr, cm^−1^) *ν*:2923 (C-H), 1737,1638 (C=O), 1561–1405 (C=C), 1270 (C-O); ^1^H NMR (400 MHz, CDCl_3_) δ 8.22–8.13 (m, 1H, Py-H), 7.77 (s, 1H, Py-H), 7.52 (dd, *J* = 13.4, 7.4 Hz, 2H, Ar-H), 7.44 (s, 1H, Ar-H), 7.35 (d, *J* = 7.2 Hz, 1H, Ar-H), 7.10 (t, *J* = 5.2 Hz, 1H, Py-H), 4.82 (s, 1H, N-CH), 3.71 (s, 3H, OCH_3_), 3.25–3.02 (m, 2H, S-CH_2_), 2.12–1.31 (m, 10 H, -C-CH_3_); ^13^C NMR (101 MHz, CDCl_3_) δ 170.56, 164.64, 153.53, 148.17, 130.20, 124.66, 121.85, 119.42, 118.32, 67.09, 52.91, 36.02, 34.43, 30.81, 25.48, 25.41, 24.54; HRMS calcd. for [M + H^+^] C_23_H_24_F_3_N_2_O4_S_: 481.1403, found 481.1408.

*(2,2-diethylthiazolidin-3-yl)(2-(3-(trifluoromethyl)phenoxy)pyridin-3-yl)methanone**(***4e***)* White solid, m.p. 80–81 °C; IR (KBr, cm^−1^) *ν*: 2954 (C-H), 1630 (C=O), 1566–1410 (C=C), 1245 (C-O); ^1^H NMR (400 MHz, CDCl_3_) δ 8.19 (dd, *J* = 4.9, 1.9 Hz, 1H, Py-H), 7.75 (dd, *J* = 7.3, 2.0 Hz, 1H, Py-H), 7.59–7.47 (m, 2H, Ar-H), 7.45 (t, *J* = 1.9 Hz, 1H, Ar-H), 7.38–7.34 (m, 1H, Ar-H), 7.13 (dd, *J* = 7.3, 4.9 Hz, 1H, Py-H), 3.89 (d, *J* = 75.8 Hz, 2H, N-CH_2_), 2.91 (t, *J* = 6.1 Hz, 2H, S-CH_2_), 2.63–2.03 (m, 4H, -CH_2_-), 1.10 (t, *J* = 7.3 Hz, 6H,-C-CH_3_); ^13^C NMR (101 MHz, CDCl_3_) δ 163.72, 157.44, 153.34, 147.97, 137.51, 130.18, 124.73, 123.79, 121.78, 121.74, 119.41, 118.42, 118.38, 81.50, 55.48, 31.69, 28.81, 9.21; HRMS calcd; for [M + H^+^] C_20_H_22_F_3_N_2_O_2_S: 411.1349, found 411.1347.

*Methyl (R)-2,2-diethyl-3-(2-(3-(trifluoromethyl)phenoxy)nicotinoyl)thiazolidine-4-carboxylate**(***4f***)* White solid, m.p. 141–142 °C; IR (KBr, cm^−1^) *ν*: 2928 (C-H), 1728, 1633 (C=O), 1566–1408 (C=C), 1272 (C-O); ^1^H NMR (400 MHz, CDCl_3_) δ 8.17 (dd, *J* = 4.9, 1.9 Hz, 1H, Py-H), 7.82 (d, *J* = 6.8 Hz, 1H, Py-H), 7.60–7.48 (m, 2H, Ar-H), 7.44 (s, 1H, Ar-H), 7.36 (d, *J* = 7.7 Hz, 1H, Ar-H), 7.10 (dd, *J* = 7.4, 4.9 Hz, 1H, Py-H), 4.85 (dd, *J* = 6.1, 2.2 Hz, 1H, N-CH), 3.67 (s, 3H, O-CH_3_), 3.38–3.12 (m, 2H S-CH_2_), 2.65–2.02 (m, 4H, -CH_2_-), 1.08 (td, *J* = 7.4, 4.8 Hz, 6H, -C-CH_3_); ^13^C NMR (101 MHz, CDCl_3_) δ 170.49, 164.65, 153.21, 148.24, 130.24, 124.75, 123.04, 121.94, 121.90, 119.30, 118.45, 83.69, 66.88, 52.80, 32.47, 31.53, 30.75, 10.32, 8.90; HRMS calcd. for [M + H^+^] C_22_H_24_F_3_N_2_O_4_S: 469.1403, found 469.1406.

*(2-(4-chlorophenoxy)pyridin-3-yl)(2-methyl-2-propylthiazolidin-3-yl)methanone**(***4g***)* White solid, m.p. 75–76 °C; IR (KBr, cm^−1^) *ν*: 3029–2846 (C-H), 1614 (C=O), 1576–1482 (C=C), 1248 (C-O); ^1^H NMR (400 MHz, CDCl_3_) δ 8.17 (d, *J* = 4.7 Hz, 1H, Py-H), 7.72 (d, *J* = 7.0 Hz, 1H, Py-H), 7.36 (d, *J* = 8.3 Hz, 2H, Ar-H), 7.09 (dd, *J* = 13.4, 6.6 Hz, 3H, Ar-H, Py-H), 3.84 (s, 2H, N-CH_2_), 3.04 -2.80 (m, 2H, S-CH_2_), 2.47–1.93 (m, 4H, C-CH_2_-CH_2_-), 1.82–1.26 (t, 3H, -C-CH_3_), 0.94 (s, 3H, -CH_3_); ^13^C NMR (101 MHz, CDCl_3_) δ 163.94, 157.89, 151.83, 148.06, 137.55, 130.27, 129.66, 123.67, 122.60, 119.11, 54.74, 28.57, 18.58, 14.07; HRMS calcd. for [M + H^+^] C_19_H_22_ClN_2_O_2_S: 377.1085, found 377.1085.

*(1-thia-4-azaspiro*[4,4]*nonan-4-yl)(2-(3-(trifluoromethyl)phenoxy)pyridin-3-yl)methanone**(***4h***)* White solid, m.p. 102–103 °C; IR (KBr, cm^−1^) *ν*: 2912 (C-H), 1619 (C=O), 1571–1391 (C=C), 1237 (C-O); ^1^H NMR (400 MHz, CDCl_3_) δ 8.20 (dd, *J* = 4.9, 1.9 Hz, 1H, Py-H), 7.77 (dd, *J* = 7.4, 1.9 Hz, 1H, Py-H), 7.53 (dt, *J* = 14.7, 7.8 Hz, 2H, Ar-H), 7.46 (s, 1H, Ar-H), 7.37 (d, *J* = 7.6 Hz, 1H, Ar-H), 7.13 (dd, *J* = 7.4, 4.9 Hz, 1H, Py-H), 3.88 (s, 2H, N-CH_2_), 2.98 (t, *J* = 6.0 Hz, 2H, S-CH_2_), 2.96–2.01 (m, 4H, -CH_2_-), 1.91–1.66 (m, 4H, -CH_2_-); ^13^C NMR (101 MHz, CDCl_3_) δ 163.54, 157.79, 153.59, 148.10, 137.55, 130.12, 124.83, 123.69, 121.76, 121.72, 119.36, 118.54, 118.50, 81.32, 54.14, 38.57, 29.08, 25.26; HRMS calcd. for [M + H^+^] C_20_H_20_F_3_N_2_O_2_S: 409.1192, found 409.1194.

*(1-thia-4-azaspiro*[4,5]*decan-4-yl)(2-(3-(trifluoromethyl)phenoxy)pyridin-3-yl)methanon*e *(**4i**)* White solid, m.p. 135–136 °C; IR (KBr, cm^−1^) *ν*: 2908 (C-H), 1624 (C=O), 1531–1441 (C=C), 1228 (C-O); ^1^H NMR (400 MHz, CDCl_3_) δ 8.19 (dd, *J* = 4.9, 1.9 Hz, 1H, Py-H),7.76 (dd, *J* = 7.4, 1.9 Hz, 1H, Py-H), 7.52 (dt, *J* = 15.2, 7.9 Hz, 2H, Ar-H), 7.45 (d, *J* =1.8 Hz, 1H, Ar-H), 7.36 (dt, *J* = 7.9, 1.8 Hz, 1H, Ar-H), 7.13 (dd, *J* = 7.4, 5.0 Hz, 1H, Py-H), 3.92 (d, *J* = 51.7 Hz, 2H, NCH_2_), 3.16–2.99 (m, 2H, S-CH_2_), 2.90–1.26 (m, 10 H, -C_5_H_10_); ^13^CNMR (101 MHz, CDCl_3_) δ 164.10, 157.64, 153.59, 147.97, 137.46, 130.14, 124.76, 124.13, 121.74, 121.70, 119.43, 118.44, 118.40, 54.55, 35.53, 27.94, 25.31, 24.60; HRMS calcd. for [M + H^+^] C_21_H_22_F_3_N_2_O_2_S: 423.1349, found 423.1346.

*(1-methyl-2-propylthiazolidin-3-yl)(2-(3-(trifluoromethyl)phenoxy)pyridin-3-yl)methanone**(***4j***)* White solid, m.p. 89–90 °C; IR (KBr, cm^−1^) *ν*: 2908 (C-H), 1624 (C=O), 1561–1441 (C=C), 1228 (C-O); ^1^H NMR (400 MHz, CDCl_3_) δ 8.18 (dd, *J* = 5.0, 1.8 Hz, 1H, Py-H), 7.75 (d, *J* = 6.3 Hz, 1H, Py-H), 7.51 (dt, *J* = 16.7, 7.9 Hz, 2H, Ar-H), 7.45 (s, 1H, Ar-H), 7.41–7.34 (m, 1H, Ar-H), 7.12 (dd, *J* = 7.3, 4.9 Hz, 1H, Py-H), 3.87 (s, 2H, N-CH_2_), 3.12–2.74 (m, 2H, S-CH_2_), 2.48–1.62 (m, 7H, -CH_2_-CH_2_-CH_3_), 0.95 (s, *J* = 2.4 Hz, 3H, -CH_3_); ^13^C NMR (101 MHz, CDCl_3_) δ 163.77, 157.50, 153.51, 148.01, 137.58, 130.14, 124.60, 123.77, 121.67, 121.63, 119.45, 118.31, 76.30, 54.75, 41.72, 28.51, 18.58, 14.02; HRMS calcd. for [M + H^+^] C_20_H_22_F_3_N_2_O_2_S: 411.1349, found 411.1348.

*(2-(4-chloro-3,5-dimethylphenoxy)pyridin-3-yl)(1-thia-4-azaspiro[4.5]decan-4-yl)methanone**(***4k***)* White solid, m.p. 149–150 °C; IR (KBr, cm^−1^) *ν*: 2904 (C-H), 1618 (C=O), 1563–1391 (C=C), 1242 (C-O); ^1^H NMR (400 MHz, CDCl_3_) δ 8.19 (dd, *J* = 5.0, 1.9 Hz, 1H, Py-H), 7.73 (dd, *J* = 7.3, 1.9 Hz, 1H, Py-H), 7.07 (dd, *J* = 7.3, 4.9 Hz, 1H, Ar-H), 6.90 (s, 2H, Ar-H, Py-H), 3.92 (d, *J* = 77.0 Hz, 2H, N-CH_2_), 3.09 (s, 2H, S-CH_2_), 2.89 (t, *J* = 5.9 Hz, 2H, -CH_2_), 2.39 (s, 6H, Ar-CH_3_), 1.90–1.32 (m, 8H, -C_4_H_8_); ^13^C NMR (101 MHz, CDCl_3_) δ 164.38, 158.31, 151.03, 148.09, 137.64, 137.28, 123.96, 121.18, 118.77, 79.66, 54.47, 27.97, 25.33, 24.62, 20.93; HRMS calcd. for [M + H^+^] C_22_H_26_ClN_2_O_2_S: 417.1398, found 417.1393.

*(2-(4-chloro-3,5-dimethylphenoxy)pyridin-3-yl)(2,2-diethylthiazolidin-3-yl)methanone**(***4l***)* White solid, m.p. 129–130 °C; IR (KBr, cm^−1^) *ν*: 2908 (C-H), 1627 (C=O), 1562–1411 (C=C), 1239 (C-O); ^1^H NMR (400 MHz, CDCl_3_) δ 8.20 (dd, *J* = 5.0, 1.8 Hz, 1H, Py-H), 7.72 (dd, *J* = 7.4, 1.8 Hz, 1H, Py-H), 7.08 (dd, *J* = 7.3, 4.9 Hz, 1H, Ar-H), 6.90 (s, 2H, Ar-H, Py-H), 3.90 (d, *J* = 97.4 Hz, 2H, N-CH_2_), 2.91 (s, 2H, S-CH_2_), 2.63 (s, 2H, -CH_2_-), 2.40 (s, 6H, Ar-CH_3_), 1.45 (s, 2H, -CH_2_-), 1.10 (t, *J* = 7.3 Hz, 6H, -C-CH_3_); ^13^C NMR (101 MHz, CDCl_3_) δ 164.01, 158.17, 150.76, 148.14, 137.69, 137.38, 131.08, 123.60, 121.21, 118.74, 81.49, 55.42, 28.87, 26.92, 20.97; HRMS calcd. for [M + H^+^] C_21_H_26_ClN_2_O_2_S: 405.1398, found 405.1396.

*(2-(4-chloro-3,5-dimethylphenoxy)pyridin-3-yl)(2,2-dipropylthiazolidin-3-yl)methanone**(***4m***)* White solid, m.p. 135–136 °C; IR (KBr, cm^−1^) *ν*: 2934 (C-H), 1613 (C=O), 1562–1409 (C=C), 1236 (C-O); ^1^H NMR (400 MHz, CDCl_3_) δ 8.20 (d, *J* = 3.1 Hz, 1H, Py-H), 7.71 (s, 1H, Py-H), 7.29 (s, 1H, Ar-H), 6.90 (s, 2H, Ar-H, Py-H), 3.87 (d, *J* = 95.3 Hz, 2H, N-CH_2_), 2.90 (t, *J* = 6.0 Hz, 2H, S-CH_2_), 2.52 (t, *J* = 3.9 Hz, 2H, -CH_2_-), 2.40 (s, 6H, Ar-CH_3_), 1.94 (t, *J* = 39.7 Hz, 2H, -CH_2_-), 1.64 (s, 4H, -C-CH_2_-), 0.96 (d, *J* = 28.9 Hz, 6H,-C-CH_3_); ^13^C NMR (101 MHz, CDCl_3_) δ 163.94, 158.08, 148.12, 137.63, 137.45, 123.69, 121.02, 118.78, 80.12, 55.16, 41.09, 28.93, 20.97, 14.15, 14.15; HRMS calcd. for [M + H^+^] C_23_H_30_ClN_2_O_2_S: 433.1711, found 433.1712.

*(2-(4-chloro-3,5-dimethylphenoxy)pyridin-3-yl)(2,2-dimethylthiazolidin-3-yl)methanone**(***4n***)* White solid, m.p. 124–126 °C; IR (KBr, cm^−1^) *ν*: 2956 (C-H), 1619 (C=O), 1563–1441 (C=C), 1236 (C-O); ^1^H NMR (400 MHz, CDCl_3_) δ 8.20 (dd, *J* = 5.0, 1.9 Hz, 1H, Py-H), 7.74 (dd, *J* = 7.4, 1.8 Hz, 1H, Py-H), 7.08 (dd, *J* = 7.3, 4.9 Hz, 1H, Ar-H), 6.91 (s, 2H, Ar-H, Py-H), 4.20–3.89 (m, 2H, N-CH_2_), 3.00 (s, 2H, S-CH_2_), 2.40–1.99 (m, 6H, -(CH_3_)_2_); ^13^C NMR (101 MHz, CDCl_3_) δ 164.20, 158.37, 150.98, 148.22, 137.66, 137.45, 131.02, 123.42, 121.21, 118.76, 72.26, 60.40, 54.31, 28.50, 20.95, 14.21; HRMS calcd. for [M + H^+^] C_19_H_22_Cl N_2_O_2_S: 377.1085, found 377.1085.

*(2-(tert-butyl)-2-methylthiazolidin-3-yl)(2-(4-chloro-3,5-dimethylphenoxy)pyridin-3-yl)methanone**(***4o***)* White solid, m.p. 112–114 °C; IR (KBr, cm^−1^) *ν*: 3032–2849 (C-H), 1623 (C=O), 1563–1416 (C=C), 1232 (C-O); ^1^H NMR (400 MHz, CDCl_3_) δ 8.20 (dd, *J* = 4.9, 1.9 Hz, 1H, Py-H), 7.79–7.62 (m, 1H, Py-H), 7.08 (dd, *J* = 7.3, 4.9 Hz, 1H, Ar-H), 6.91 (s, 2H, Ar-H, Py-H), 3.89 (d, *J* = 64.2 Hz, 2H, N-CH_2_), 3.17–2.83 (m, 2H, S-CH_2_), 2.52–0.96 (m, 12H, -CH_3_); ^13^C NMR (101 MHz, CDCl_3_) δ 164.07, 158.24, 150.92, 148.18, 137.64, 137.45, 123.57, 121.11, 118.76, 54.67, 28.59, 20.95, 18.58, 14.09; HRMS calcd. for [M + H^+^] C_22_H_28_ClN_2_O_2_S: 419.1555, found 419.1558.

*(2-(4-chlorophenoxy)pyridin-3-yl)(1-thia-4-azaspiro[4.4]nonan-4-yl)methanone**(***4p***)* White solid, m.p. 127–129 °C; IR (KBr, cm^−1^) *ν*: 2909 (C-H), 1630 (C=O), 1564–1475 (C=C), 1249 (C-O); ^1^H NMR (400 MHz, CDCl_3_) δ 8.32–7.92 (m, 1H, Py-H), 7.93 (d, *J* = 7.9 Hz, 1H, Py-H), 7.72 (dd, *J* = 7.4, 1.9 Hz, 1H, Ar-H), 7.47–7.30 (m, 2H, Ar-H), 7.19–6.99 (m, 3H, Ar-H, Py-H), 3.84 (s, 2H, N-CH_2_), 2.94 (t, *J* = 6.0 Hz, 2H, S-CH_2_), 2.92–1.58 (m, 8H, -C_4_H_8_); ^13^C NMR (101 MHz, CDCl_3_) δ 163.69, 151.96, 148.08, 137.48, 129.64, 123.63, 122.71, 119.05, 81.32, 54.10, 38.58, 29.07, 25.24; HRMS calcd. for [M + H^+^] C_19_H_20_ClN_2_O_2_S: 375.0929, found 375.0925.

*(2-(4-chlorophenoxy)pyridin-3-yl)(1-thia-4-azaspiro[4.5]decan-4-yl)methanone**(***4q***)* White solid, m.p. 135–136 °C; IR (KBr, cm^−1^) *ν*: 2906 (C-H), 1613 (C=O), 1565–1475 (C=C), 1244 (C-O); ^1^H NMR (400 MHz, CDCl_3_) δ 8.16 (dd, *J* = 5.0, 1.7 Hz, 1H, Py-H), 7.72 (dd, *J* = 7.3, 1.8 Hz, 1H, Py-H), 7.36 (d, *J* = 8.6 Hz, 2H, Ar-H), 7.10 (d, *J* = 8.8 Hz, 3H, Ar-H, Py-H), 3.88 (d, *J* = 56.0 Hz, 2H, N-CH_2_), 3.07 (s, 2H, S-CH_2_), 2.86–1.00 (m, 10H, -C_5_H_10_); ^13^C NMR (101 MHz, CDCl_3_) δ 157.95, 151.95, 147.98, 137.39, 130.25, 129.65, 124.07, 122.65, 119.12, 54.54, 27.95, 25.32, 24.62; HRMS calcd. for [M + H^+^] C_20_H_22_ClN_2_O_2_S: 389.1085, found 389.1088.

*(2-(4-chlorophenoxy)pyridin-3-yl)(2,2-dipropylthiazolidin-3-yl)methanone**(***4r***)* White solid, m.p. 112–113 °C; IR (KBr, cm^−1^) *ν*: 2933 (C-H), 1623 (C=O), 1563–1416 (C=C), 1233 (C-O); ^1^H NMR (400 MHz, CDCl_3_) δ 8.15–8.06 (m, 1H, Py-H), 7.71 (dd, *J* = 7.3, 1.8 Hz, 1H, Py-H), 7.49 (d, *J* = 2.4 Hz, 1H, Ar-H), 7.29 (dd, *J* = 8.6, 2.5 Hz, 2H, Ar-H), 7.16 (d, *J* = 8.7 Hz, 1H, Ar-H), 7.07 (dd, *J* = 7.3, 4.9 Hz, 1H, Py-H), 3.91 (d, *J* = 152.2 Hz, 2H, N-CH_2_), 2.87 (d, *J* = 6.1 Hz, 2H, S-CH_2_), 2.50–1.94 (m, 4H, -CH_2_-), 1.66–1.47 (m, 4H, -C-CH_2_-), 0.94 (s, 6H, -C-CH_3_); ^13^C NMR (101 MHz, CDCl_3_) δ 164.05, 158.23, 150.93, 148.16, 137.65, 130.98, 123.59, 121.11, 118.78, 54.69, 28.59, 20.95, 18.58, 14.09; HRMS calcd. for [M + H^+^] C_21_H_26_ClN_2_O_2_S: 405.1398, found 405.1395.

*(2-(4-chlorophenoxy)pyridin-3-yl)(2,2-dimethylthiazolidin-3-yl)methanone**(***4s***)* White solid, m.p. 117–118 °C; IR (KBr, cm^−1^) *ν*: 2958 (C-H), 1621 (C=O), 1551–1476 (C=C), 1245 (C-O); ^1^H NMR (400 MHz, CDCl_3_) δ 8.17 (dd, *J* = 5.0, 1.9 Hz, 1H, Py-H), 7.73 (dd, *J* = 7.3, 1.9 Hz, 1H, Py-H), 7.36 (d, *J* = 8.9 Hz, 2H, Ar-H), 7.20–7.00 (m, 3H, Ar-H, Py-H), 3.86 (s, 2H, N-CH_2_), 2.97 (t, *J* = 6.0 Hz, 2H,S-CH_2_), 1.97 (s, 6H, -(CH_3_)_2_); ^13^C NMR (101 MHz, CDCl_3_) δ 164.08, 158.04, 151.92, 148.10, 137.48, 130.30, 129.65, 123.52, 122.66, 119.07, 72.28, 54.31, 28.89, 28.46; HRMS calcd. for [M + H^+^] C_17_H_18_ClN_2_O_2_S: 349.0772, found 349.0778.

*(2-(4-chlorophenoxy)pyridin-3-yl)(2,2-diethylthiazolidin-3-yl)methanone**(***4t***)* White solid, m.p. 91–92 °C; IR (KBr, cm^−1^) *ν*: 2944 (C-H), 1624 (C=O), 1581–1475 (C=C), 1243 (C-O); ^1^H NMR (400 MHz, CDCl_3_) δ 8.16 (dd, *J* = 4.9, 1.9 Hz, 1H, Py-H), 7.71 (dd, *J* = 7.3, 1.9 Hz, 1H, Py-H), 7.36 (d, *J* = 8.8 Hz, 2H, Ar-H), 7.15–6.96 (m, 3H, Ar-H, Py-H), 3.86 (d, *J* = 84.5 Hz, 2H, N-CH_2_), 2.87 (d, *J* = 6.2 Hz, 2H, S-CH_2_), 2.61–1.97 (m, 4H, -CH_2_-), 1.08 (t, *J* = 7.3 Hz, 6H, -C-CH_3_); ^13^C NMR (101 MHz, CDCl_3_) δ 163.88, 157.77, 151.67, 147.98, 137.43, 130.33, 123.71, 122.64, 119.07, 81.51, 55.43, 28.83; HRMS calcd. for [M + H^+^] C_2__2_H_2__6_F_3_N_2_O_2_S: 439.1662, found 439.1665.

*(2-(tert-butyl)-2-methylthiazolidin-3-yl)(2-(4-chlorophenoxy)pyridin-3-yl)methanone**(***4u***)* White solid, m.p. 86–88 °C; IR (KBr, cm^−1^) *ν*: 2936–2848 (C-H), 1624 (C=O), 1581–1475 (C=C), 1240 (C-O); ^1^H NMR (400 MHz, CDCl_3_) δ 8.16 (dd, *J* = 4.9, 1.9 Hz, 1H, Py-H), 7.72 (d, *J* = 5.5 Hz, 1H, Py-H), 7.36 (d, *J* = 8.8 Hz, 2H, Ar-H), 7.13–7.05 (m, 3H, Ar-H, Py-H), 3.85 (s, 2H, N-CH_2_), 3.11–2.81 (m, 2H, S-CH_2_), 2.47–0.66 (m, 12H, -CH_3,_ -(CH_3_)_3_); ^13^C NMR (101 MHz, CDCl_3_) δ 163.94, 157.88, 151.84, 148.04, 137.51, 130.24, 129.64, 123.67, 122.57, 119.09, 54.70, 28.55, 18.57, 14.07; HRMS calcd. for [M + H^+^] C_20_H_24_ClN_2_O_2_S: 391.1242, found 391.1244.

*(2-(2,4-dichlorophenoxy)pyridin-3-yl)(2,2-dimethylthiazolidin-3-yl)methanone**(***4v***)* White solid, m.p. 121–122 °C; IR (KBr, cm^−1^) *ν*: 3029–2966(C-H), 1627 (C=O), 1559–1466 (C=C), 1253 (C-O); ^1^H NMR (400 MHz, CDCl_3_) δ 8.22–7.99 (m, 1H, Py-H), 7.89–7.66 (m, 1H, Py-H), 7.48 (d, *J* = 2.4 Hz, 1H, Ar-H), 7.29 (dd, *J* = 8.7, 2.4 Hz, 1H, Ar-H), 7.19 (d, *J* = 8.7 Hz, 1H, Ar-H), 7.12–7.03 (m, 1H, Py-H), 3.94 (s, 2H, N-CH_2_), 2.98 (t, *J* = 5.9 Hz, 2H S-CH_2_), 1.98 (s, 6H, -(CH_3_)_2_); ^13^C NMR (101 MHz, CDCl_3_) δ 163.79, 157.20, 147.92, 147.88, 137.45, 131.16, 130.23, 128.13, 127.97, 124.86, 122.98, 119.16, 72.28, 54.55, 28.72, 28.49; HRMS calcd. for [M + H^+^] C_17_H_17_Cl_2_N_2_O_2_S: 383.0382, found 383.0386.

*(2-(2,4-dichlorophenoxy)pyridin-3-yl)(2,2-diethylthiazolidin-3-yl)methanon**(***4w***)* White solid, m.p. 75–77 °C; IR (KBr, cm^−1^) *ν*: 2948 (C-H), 1627 (C=O), 1578–1464 (C=C), 1253 (C-O); ^1^H NMR (400 MHz, CDCl_3_) δ 8.12 (dd, *J* = 5.1, 1.8 Hz, 1H, Py-H), 7.71 (dd, *J* = 7.3, 1.8 Hz, 1H, Py-H), 7.49 (d, *J* = 2.4 Hz, 1H, Ar-H), 7.33–7.27 (m, 1H, Ar-H), 7.16 (d, *J* = 8.6 Hz, 1H, Ar-H), 7.08 (dd, *J* = 7.3, 4.9 Hz, 1H, Py-H), 3.93 (d, *J* = 158.6 Hz, 2H, N-CH_2_), 2.89 (s, 2H, S-CH_2_), 2.62–1.97 (m, 4H, -CH_2_-), 1.08 (t, *J* = 7.2 Hz, 6H,-C-CH_3_); ^13^C NMR (101 MHz, CDCl_3_) δ 163.60, 157.07, 147.84, 147.78, 137.41, 131.32, 130.29, 128.24, 128.03, 124.97, 123.08, 119.13, 81.45, 55.73, 32.29, 28.89; HRMS calcd. for [M + H^+^] C_19_H_21_Cl_2_N_2_O_2_S: 411.0695, found 411.0695.

*(2-(2,4-dichlorophenoxy)pyridin-3-yl)(1-thia-4-azaspiro[4.4]nonan-4-yl)methanone**(***4x***)* White solid, m.p. 95–97 °C; IR (KBr, cm^−1^) *ν*: 3034–2929 (C-H), 1626 (C=O), 1556–1466 (C=C), 1253 (C-O); ^1^H NMR (400 MHz, CDCl_3_) δ 8.12 (dd, *J* = 4.9, 1.7 Hz, 1H, Py-H), 7.72 (d, *J* = 7.3 Hz, 1H, Py-H), 7.48 (d, *J* = 2.4 Hz, 1H, Ar-H), 7.29 (dd, *J* = 8.7, 2.4 Hz, 1H, Ar-H), 7.20 (d, *J* = 8.7 Hz, 1H, Ar-H), 7.08 (dd, *J* = 7.3, 5.0 Hz, 1H, Py-H), 3.91 (s, 2H, N-CH_2_), 2.95 (t, *J* = 6.0 Hz, 2H, S-CH_2_), 2.90–1.95 (m, 4H, -CH_2_-), 1.87–1.63 (m, 4H, -CH_2_-); ^13^C NMR (101 MHz, CDCl_3_) δ 163.41, 157.25, 147.97, 147.84, 137.36, 131.11, 130.20, 128.07, 127.98, 124.86, 119.11, 81.33, 54.31, 38.61, 29.07, 25.24; HRMS calcd. for [M + H^+^] C_19_H_19_Cl_2_N_2_O_2_S: 409.0539, found 409.0533.

*2-(2,4-dichlorophenoxy)pyridin-3-yl)(2,2-dipropylthiazolidin-3-yl)methanone**(***4y***)* White solid, m.p. 112–114 °C; IR (KBr, cm^−1^) *ν*: 3075–2847 (C-H), 1630 (C=O), 1563–1463 (C=C), 1250 (C-O); ^1^H NMR (400 MHz, CDCl_3_) δ 8.11 (d, *J* = 3.3 Hz, 1H, Py-H), 7.71 (d, *J* = 6.2 Hz, 1H, Py-H), 7.49 (d, *J* = 2.4 Hz, 1H, Ar-H), 7.29 (dd, *J* = 8.7, 2.4 Hz, 1H, Ar-H), 7.16 (d, *J* = 8.7 Hz, 1H, Ar-H), 7.07 (dd, *J* = 7.1, 5.0 Hz, 1H, Py-H), 3.91 (d, *J* = 152.4 Hz, 2H, N-CH_2_), 2.88 (t, *J* = 6.0 Hz, 2H, S-CH_2_), 2.50–1.94 (m, 4H, -CH_2_-), 1.66–1.47 (s, 4H, -C-CH_2_-), 0.94 (s, 6H, -C-CH_3_); ^13^C NMR (101 MHz, CDCl_3_) δ 163.53, 157.04, 147.85, 137.53, 131.23, 130.28, 128.21, 127.99, 124.82, 123.16, 119.15, 80.11, 55.44, 28.96, 14.13; HRMS calcd. for [M + H^+^] C_21_H_25_Cl_2_N_2_O_2_S: 439.1008, found 439.1010.

*(2-(4-chloro-3,5-dimethylphenoxy)pyridin-3-yl)(1-thia-4-azaspiro[4.4]nonan-4-yl)methanone**(***4z***)* White solid, m.p. 117–119 °C; IR (KBr, cm^−1^) *ν*: 2908 (C-H), 1631 (C=O), 1566–1465 (C=C), 1253 (C-O); ^1^H NMR (400 MHz, CDCl_3_) δ 8.18 (dd, *J* = 4.9, 1.9 Hz, 1H, Py-H), 7.71 (dd, *J* = 7.3, 2.0 Hz, 1H, Py-H), 7.05 (dd, *J* = 7.3, 5.0 Hz, 1H, Ar-H), 6.88 (s, 2H, Ar-H, Py-H), 4.21–3.84 (m, 2H, N-CH_2_), 2.95 (t, *J* = 6.0 Hz, 2H, S-CH_2_), 2.37–1.26 (m, 12 H, -CH_3_, -(CH_3_)_3_); ^13^C NMR (101 MHz, CDCl_3_) δ 163.83, 151.05, 148.22, 137.62, 137.35, 123.54, 121.25, 118.70, 81.29, 60.38, 54.05, 38.60, 29.09, 25.26, 20.93, 14.21; HRMS calcd. for [M + H^+^] C_21_H_24_ClN_2_O_2_S: 403.1242, found 403.1242.

IR, NMR and HRMS spectra of compound **4a–4z** can be found in the Appendix A.

### 2.3. X-ray Diffraction

A single crystal of compound **4a** was recrystallized from EtOAc and n-hexane under appropriate conditions. A D-8 VENTURE X-diffractometer (Bruker) was used to obtain X-ray data, which were deposited at the Cambridge Crystallographic Data Centre (CCDC no. 1856972). Crystal information of compound **4a** can be found in the Appendix A.

### 2.4. Plant Material and Growth Conditions

After an incipient screening experiment, the optimal concentration of compound **4** for the bioassay was confirmed prior to testing. Compound **4** was tested for safener activity for protection of maize from fomesafen (0.8 mg·kg^−1^). Seeds were soaked in a solution of thiazole phenoxypyridines or water overnight. Afterwards, germination was performed to obtain seeds with the same germination status. The selected seeds were sowed in pots that contained soil mixed with a fomesafen solution (0.8 mg·kg^−1^), and the control was treated with water. All pots were incubated in an illumination box. Plant material was harvested 7 d after the treatment began. The plants were washed, and the water was drained. The shoots and roots of the plants were separated and examined. The growth index recovery rates were calculated using the following formula [12]
$$\mathrm{Recovery}\;\mathrm{rate}\;\left(\%\right)=\frac{\mathrm{Treated}\;\mathrm{with}\;\mathrm{compounds}-\mathrm{Treated}\;\mathrm{with}\;\mathrm{herbicide}}{\;\mathrm{Contrast}\;-\mathrm{Treated}\;\mathrm{with}\;\mathrm{herbicide}}\times 100$$

The shoots and roots were stored in an ultralow-temperature refrigerator at −80 °C for biological activity assays (glutathione, GSH; glutathione *S*-transferase, GST; PPO). The experiment was repeated three times.

### 2.5. GSH Content Assay

The GSH content in maize roots treated with different compounds and the control was determined. GSH was extracted from 0.2 g of fresh root. After flash freezing in liquid nitrogen, the maize root tissue was homogenized in 1.2 mL of trichloroacetic acid (TCA, 5% *w*/*v*) at a low temperature. The supernatant was aspirated immediately after centrifugation (15,000× *g*, 20 min) and was added to 1.6 mL of phosphate buffer (PB, 0.5 M; pH 8.0). Then, 16 μL of 5,5′-dithiobis-(2-nitrobenzoic acid) (DTNB, 10 mM) reagent was added. The absorbance at 412 nm was recorded on a spectrophotometer, and the concentration was calculated via comparison with a known concentration [31].

### 2.6. GST Activity Assay

The specific activity of GST was determined by a modified version of the method described by Scarponi [32]. Maize root (0.2 g) was used to determine GST activity. Roots were flash frozen in liquid nitrogen and ground into a powder, and 1 mL of enzyme extraction solution (PB, 100 mM, pH 7.8; with sodium pyrosulfite (1 mM) and polyvinyl pyrrolidone (5%, *w*/*v*)). After centrifugation at 15,000× *g* for 20 min, the supernatant was removed. Root extract (25 μL), GSH (100 mM, pH 7.0, 50 μL), PB (100 mM, pH 6.5, 0.9 mL), and chlorodinitrobenzene (CDNB, 20 mM dissolved in 96% ethanol) were sequentially added to the reactor. The reaction was monitored at 340 nm for 5 min, and the absorbance of the solution was measured at 60 s intervals. The level of the conjugate composed of GSH and CDNB formed per unit of time per mg of enzyme (nmol·s^−1^·mg^−1^ protein) was measured as an indicator of GST activity.

### 2.7. PPO Activity Assay

The effect of thiazole phenoxypyridines on the PPO activity to protect maize from fomesafen was evaluated [33]. The maize leaves (2 g), Tris homogenization buffer (tris(hydroxymethyl)aminomethane (Tris)-HCl (50 mM, pH 7.3), bovine serum albumin (BSA, 2 g·L^−1^), sucrose (0.5 M), MgCl_2_ (1 mM) and ethylenediaminetetraacetic acid (EDTA, 1 mM)) was added in 25 mL flask. The materials were homogenized with a high-speed homogenizer, filtered through a 100 mesh gauze and centrifuged for 2 min (800× *g*, 0 °C). The supernatant was removed and centrifuged again for 6 min (17,000× *g*, 0 °C). The precipitate was dissolved in configured lysis buffer, and the obtained enzyme sample was stored in an ultralow-temperature refrigerator. PPO activity was determined using an enzyme-linked immunosorbent assay (ELISA) kit.

### 2.8. Molecular Docking

The three-dimensional structures of fomesafen and compound **4i** were modelled using SYBYL-X 2.0. Subsequently, Gasteiger–Huckel charges were also determined during optimization of the molecule. Molecular docking was performed using CDOCKER modules in Discovery Studio 2.5 (BIOVIA Inc., San Diego, CA, USA). The parameters are used in the molecular docking study with the default docking setting (Top Hits:10; Random Conformations: 10; Dynamics Steps: 1000; Forcefield: CHARMm; Ligand Partial Charge Method: Momany-Rone; Final Minimization: Full Potential). -CDOCKER ENERGY (compounds **4i**): 11.339 kcal/mol; -CDOCKER ENERGY (fomesafen): 31.0016 kcal/mol. The crystal structure of PPO (PDB ID, 1SEZ) was obtained from the Protein Data Bank. Water and other small molecules were expunged using Accelrys Discovery Studio 2.5 to reduce the influence of certain cocrystallized substances in the protein. A known ligand was used to determine the active site of PPO. The evaluation index for the binding energy of the small molecule-receptor protein complex and the strongly negative performance indicates a highly stable structure.

## 3. Results and Discussion

### 3.1. Synthesis

Elevated temperatures were tested in this study to increase the yield. The results showed that the product yields obtained after a few hours were higher than 46%. Phenoxypyridines were synthesized with 2-phenoxynicotinic acid and different types of phenols by nucleophilic substitution. The phenoxypyridine yield was affected mainly by the passivation effects of halogens the nucleophilicity of phenol. In the preparation of 2-phenoxynicotinoyl chloride, extremely high yields can be obtained by refluxing at 60 °C for 60 min. The yields of compounds **4a–4z** ranged from 21% to 93% (Table 1).

### 3.2. Crystallographic Data for Compound ***4a***

The crystallographic data for compound **4a** are as follows: C_20_H_19_F_3_N_2_O_4_S (M = 440.43 g/mol); crystal size, 0.130 × 0.120 × 0.100 mm; crystal system, orthorhombic; space group, P2_1_2_1_2_1_; *a* = 8.0896(3) Å; *b* = 12.0540(5) Å; *c* = 42.6668(14) Å; *V* = 4160.5(3) Å^3^; *Z* = 8; λ= 0.71073 Å; T = 100(2) K. The number of measured reflections was 54145, and the number of independent reflections was 9573 (*R_int_* = 0.0483). The final R indices were *R*_1_ = 0.0547 and *wR*_2_ = 0.1514. The largest diff. peak and hole were 0.499 and −0.341 e Å^−3^, respectively. As shown in Figure 2, in the structure, the dihedral angle between the phenyl (C2 to C7) and pyridyl (C8 to C12/N1) was 80.82°, and the dihedral angle between the pyridyl (C8/C9/C10/C11/C12/N1) and thiazolidine (N2/C14/S1/C15/C16) was 51.42°. A chiral carbon (C14) with an *R* configuration was observed. The packing view of compound **4a** indicated that the molecules were linked by weak intermolecular hydrogen bonds that formed C-H … O, and marked intermolecular π–π interactions were not observed for compound **4a** (Figure 3).

### 3.3. Maize Growth

Fomesafen can strongly retard the growth of maize. When the fomesafen concentration in soil was 0.8 mg·kg^−1^, the plant height, root length, plant weight, root weight of maize were inhibited to 34.2%, 32.6%, 27.9%, and 33.0% of the control respectively. Fortunately, compounds **4a–4z** could reduce this effect to varying degrees. The initial screening experiment showed that the most suitable concentration of compounds **4** was 10 mg·kg^−1^. The effect of thiazole phenoxypyridines protects maize from fomesafen, as evaluated by the recovery rate. The recovery rates of maize treated with compound **4** are shown in Table 2. The results showed that compounds **4i**, **4o**, and **4z** could increase the plant height and weight, which were effectively inhibited by fomesafen. Compound **4i** showed an average recovery rate of over 72% and had the best safener activity against fomesafen-induced damage.

### 3.4. GSH Content, GST Activity, and PPO Activity

After treatment with compounds **4a–4z**, the GSH content, GST activity and PPO activity were measured to examine the effects of compounds **4a–4z** (Table 3).

Some studies have proposed GSH-conjugation-based safener activity, in which the herbicides are oxidized, metabolized, conjugated to glutathione in the crop to increases crop tolerance. After treatment with compound, the GSH content in the plant roots increased by varying degrees, except with compounds **4e**, **4p**, **4t**, **4w**, and **4y**. Simultaneously, maize treated with compounds **4i** and **4z** had higher GSH content than those treated with other compounds and the control. After pretreatment with compounds **4i** and **4z**, the GSH content markedly increased by 167% and 147%, respectively.

GST could catalyze the conjugation between glutathione and the herbicide substrate in the plant, so enhancing the expression of GST can improve the tolerance of plant to herbicide [34]. The results showed that the GST activity of maize treated with compounds **4c**, **4d**, **4i**, **4m**, **4o**, and **4z** was significantly higher than that of the plants treated with fomesafen and the other compounds. Maize treated with compound **4i** exhibited the best GST activity, which was almost 163% of the fomesafen-treated group and twice of the control group.

The results showed that fomesafen inhibits the activity of PPO in maize, which is extremely unfavorable for plant growth. The PPO activity in the leaves of plants treated with compound **4** increased significantly, except with compound **4s**. In maize treated with compounds **4h**, **4i**, **4o**, and **4z**, the activity of PPO was higher than 151.37 U/L, reaching 93% of the control level (162.37 U/L). These target compounds could effectively induce PPO activity, which was inhibited by the herbicide fomesafen.

### 3.5. Structure–Activity Relationship

The safener activities of the thiazole phenoxypyridines were affected by the substituents. A summary of results is outlined in Figure 4. The structure–activity relationships for the substituents R^1^ and R^2^ were investigated. Compounds with cyclohexyl group (**4i**) and cyclopentyl group (**4h**) exhibited higher safener activity than that with methyl (**4c**) and ethyl (**4e**), while the compounds with propyl (**4m**) groups were less active. The effect at R^3^ on the safener activity was investigated with the compounds **4a** and **4d**. Compounds with a hydrogen atom (**4a**) exhibited higher safener activities than those with methoxycarbonyl (**4d**) groups. The compounds with a chlorine atom (**4v**, **4g**) on R^4^ and R^6^ showed a higher activity than those with a hydrogen atom (**4s**, **4b**). Compounds with trifluoromethyl (**4i**, **4j**, **4k**) groups at R^5^ gave a better safener activity than those with the methyl group and hydrogen atom. The structure–activity relationships for the substituents R^7^ was also investigated. Replacement of the methyl group (**4n**) by hydrogen atom gave **4m**, which was as active as **4n**. Compounds **4h**, **4i**, **4o**, and **4z** exhibited a relatively high degree of prevention of fomesafen-induced damage. As shown in Table 1, analysis of the structures of compounds **4h**, **4i**, and **4z** was performed and interestingly, the thiazolidine structures of these compounds contained a cyclic structure at R^1^ and R^2^. The effects of several compounds (**4a**, **4b**, **4d**, **4f**) with the *R* configuration were also investigated, but good detoxification effects were not observed. The results of the biological activity assays showed that compound **4i** was found to be the most active which has a spiro ring at the thiazolidine position and a trifluoromethyl substituent on the phenyl structure.

### 3.6. Molecular Docking Studies

Compound **4i** was chosen to carry out the molecular docking experiment because of its superior safener activity exhibited by this compound in the biological activity test. In docking studies, the binding modes of fomesafen (Figure 5A) and compound **4i** (Figure 5B) to PPO were elucidated. The docking results demonstrates the potential functional mechanism of fomesafen: the active molecule binds to the active pocket of PPO, which may inactivate the enzyme and block the entrance channel to the active site. In contrast, compound **4i** could bind to the active pocket of PPO and prevent fomesafen from binding to PPO, but small substrates required at the active site still have the opportunity to cross the channel [35,36,37,38]. So, the results indicated that fomesafen combines with GSH and the maize tolerance under fomesafen toxicity stress could be improved.

The details of the molecular docking were explored (Figure 6). Both fomesafen and compound **4i** docked to PPO at the same target active site residue (Arg98). There were two hydrogen bonds between the amino acid residue Arg98 and the fomesafen molecule. There was only a single hydrogen bond formed between the fluorine atom of compound **4i** and Arg98. These results could explain the different effect of compound **4i** and fomesafen to PPO activity.

## 4. Conclusions

A series of novel thiazole phenoxypyridine compounds were designed based on bioisosteric properties, active substructure combinations and structure–activity relationships. Most of the compounds **4** could improve maize tolerance under fomesafen toxicity stress to some extent through decreasing the binding of fomesafen to the target site and enhancing GSH-conjugation catalyzed by GST. Compound **4i** exhibited excellent results in various tests and molecular simulations. To the best of our knowledge, compound **4i** is the first compound to show excellent safener activity against fomesafen. This result could provide a potential method for the design of novel safeners to protect maize from fomesafen.

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
