# Peer review of "Novel Thiazole Phenoxypyridine Derivatives Protect Maize from Residual Pesticide Injury Caused by PPO-Inhibitor Fomesafen"

_biomolecules, 2019, doi:10.3390/biom9100514_

Round 1

Reviewer 1 Report

Some minor spelling mistakes:

-row 32 "chainsand"

-row 336 '

-row 46 "Dtudies"

-row 460 should be demonstrates

Explanation of abbreviations:

-row 68 ai?

-scheme 1 DMC? (if it is dichloromethane should be DCM)

-row 337 PB? (phosphate buffer?)

-table 2 and 3 letters near the values should be explained

The molecular docking study should be improved:

-what program of molecular docking have you used?

-what are the parameters used in the molecular docking study? (space, coordinates, number of different solutions, clustering?)

-what are the binding affinity of the compounds to the enzyme? 

In the IR spectrum, can you identify some signals given by the stretching of the ether group?

Author Response

Dear Reviewer:

Thank you for your comments concerning our manuscript entitled “Novel Thiazole Phenoxypyridines Derivatives Protect Maize from Residual Pesticide Injury Caused by PPO-inhibitor Fomesafen”. Those comments are all valuable and very helpful for revising and improving our paper, as well as the important guiding significance to our researches. We have studied the comments carefully and have made correction which we hope meet with approval. The revised part has been marked in red in the revision. The main corrections in the paper and the responds to your comments are as following:

Best regards,

Fei Ye

Response to Reviewer Comments:

Some minor spelling mistakes:

-row 32 "chainsand"

Response: This word “chainsand” has been revised to “chains and”. (Page 1: line 32)

-row 336 '

Response: This word “u’sed” has been revised to “used”. (Page 8: line 336)

-row 446 "Dtudies"

Response: This word “Dtudies” has been revised to “Studies”. (Page 14: line 462)

-row 460 should be demonstrates

Response: This word “demonstrate” has been revised to “demonstrates”. (Page 14: line 466)

Explanation of abbreviations:

-row 68 ai?

Response: ai indicates the active ingredient content. The sentence has been corrected in the revised version as following:

Fomesafen was provided by Jiangsu Fengshan Group Co., Ltd, the active ingredient content of 15% (m/m). (Page 3: line 68-69).

-scheme 1 DMC? (if it is dichloromethane should be DCM)

Response: Sorry for this mistake. The solvent is dichloromethane, DCM. This word “DMC” has been revised to “DCM”. Scheme 1 has been improved and re-uploaded in the revised version..

-row 337 PB? (phosphate buffer?)

Response: PB is phosphate buffer which has been marked in the text (Page 8: line 330).

-table 2 and 3 letters near the values should be explained.

Response: This question has been modified in the revised manuscript.

This explanation has been added to the revised article as follows:

different lowercase letters in the table display a significant difference (P < 0.05) in the tested products. (Page 12: line 408-409; Page 13: 418-419)

The molecular docking study should be improved:

-what program of molecular docking have you used?

Response: Molecular docking was carried out using CDOCKER modules in Discovery Studio 2.5 (BIOVIA Inc., San Diego, USA). This question has been modified in the revised manuscript. (Page 9: line 358-359)

-what are the parameters used in the molecular docking study? (space, coordinates, number of different solutions, clustering?)

Response: The parameters are used in the molecular docking study with the default docking setting. This question has been modified in the revised manuscript. (Page 9: line 359-360)

-what are the binding affinity of the compounds to the enzyme?

Response: CDOCKER ENERGY of compound 4i and fomesafen was calculated. -CDOCKER ENERGY (compounds 4i): 11.339 kcal/mol; -CDOCKER ENERGY (fomesafen): 31.0016 kcal/mol. This question has been modified in the revised manuscript. (Page 9: line 360-361)

In the IR spectrum, can you identify some signals given by the stretching of the ether group?

Response: Yes, some signals given by the stretching of the ether group can be identified in the fingerprint region. IR (KBr, cm-1) ν: 1300-1000 (-C-O-C-).

Reviewer 2 Report

Dear Authors,

      I have reviewed the manuscript entitled "Novel Thiazole Phenoxypyridines Derivatives Protect Maize from Residual Pesticide Injury Caused by PPO-inhibitors Fomesafen and even though I found the research to be weel presented and of great interest for the readers, there are some minor mistakes that should be corrected before publication.

I would change "inhibitors" to "inhibitor" (singular) from the title as it refers to only one inhibitor (Fomesafen). Page 1, line 32: Please separate "chainsand" to "chains and". Page 2, line 45: I believe the correct name for the herbicide is "fenoxaprop-P-ethyl" (the final letter is missing). Page 2, lines 52-53: I suggest the following modifications for a better understanding: "... active substructure combinations and bioisosterism have already been applied to design novel herbicide safeners". Page 3, line 93: I would use "reaction mixture" instead of "reactant" as it refers to a mix of reactants. Page 8, line 336: An error occurred when writing "used". Please correct it. Please use bold letters and numbers for the compounds' codes throughout the entire manuscript (page 9: lines 356 (4i instead of 4i) and 370 (4a-4z instead of 4a-4z), page 10: line 377 (4a instead of 4a). Page 13, line 443: I suggest the following rephrasing for a better clarity: "...and R6 showed a higher activity than those with a hydrogen atom..." Page 13, line 444-445: I suggest the following modification for a improved understanding of the text: "...gave a better safener activity than those with the methyl group and hydrogen atom..."  Page 14, line 456: Please correct "Dtudies" to "Studies". Page 14, line 464: I suggest the following changes: "...the results indicated that fomesafen combines with GSH..."

Author Response

Dear Reviewer:

Thank you for your comments concerning our manuscript entitled “Novel Thiazole Phenoxypyridines Derivatives Protect Maize from Residual Pesticide Injury Caused by PPO-inhibitor Fomesafen”. Those comments are all valuable and very helpful for revising and improving our paper, as well as the important guiding significance to our researches. We have studied the comments carefully and have made correction which we hope meet with approval. The revised part has been marked in red in the revision. The main corrections in the paper and the responds to your comments are as following.

Best regards,

Fei Ye

Response to Reviewer Comments

I would change "inhibitors" to "inhibitor" (singular) from the title as it refers to only one inhibitor (Fomesafen).

Response: This word “inhibitors” has been revised to “inhibitor”.

Page 1, line 32: Please separate "chainsand" to "chains and".

Response: This word “inhibitors” has been revised to “inhibitor” (Page 1: line 32).

Page 2, line 45: I believe the correct name for the herbicide is "fenoxaprop-P-ethyl" (the final letter is missing).

Response: This word “fenoxaprop-P-ethy” has been revised to “fenoxaprop-P-ethyl” (Page 2: line 45).

Page 2, lines 52-53: I suggest the following modifications for a better understanding: "... active substructure combinations and bioisosterism have already been applied to design novel herbicide safeners".

Response: It has been corrected in the revised manuscript (Page 2: line 52-53).

Page 3, line 93: I would use "reaction mixture" instead of "reactant" as it refers to a mix of reactants.

Response: It has been corrected in the revised manuscript (Page 3 line 93).

Page 8, line 336: An error occurred when writing "used". Please correct it.

Response: This word “u’sed” has been revised to “used” (Page 8: line 336).

Please use bold letters and numbers for the compounds' codes throughout the entire manuscript (page 9: lines 356 (4i instead of 4i) and 370 (4a-4z instead of 4a-4z), page 10: line 377 (4a instead of 4a).

Response: This word “4i” has been revised to “4i” (page 9: line 356). This word “4a-4z” has been revised to “4a-4z” (Page 9: line 373). This word “4a” has been revised to “4a” (Page 10: line 380).

Page 13, line 443: I suggest the following rephrasing for a better clarity: "...and R6 showed a higher activity than those with a hydrogen atom..."

Response: It has been corrected in the revised manuscript (Page 13: line 448).

Page 13, line 444-445: I suggest the following modification for a improved understanding of the text: "...gave a better safener activity than those with the methyl group and hydrogen atom..." 

Response: It has been corrected in the revised manuscript (Page 13 line 449-450).

Page 14, line 456: Please correct "Dtudies" to "Studies".

Response: This word “Dtudies” has been revised to “Studies” (Page 14: line 461).

Page 14, line 464: I suggest the following changes: "...the results indicated that fomesafen combines with GSH..." 

Response: It has been corrected in the revised manuscript (Page 13 line 469).

Round 2

Reviewer 1 Report

I would like to see more precise data about the parameters in the molecular docking study and some signals and values of signals given by the stretching of the ether group.

Author Response

Dear Reviewer:

Thank you for your comments concerning our manuscript entitled “Novel Thiazole Phenoxypyridines Derivatives Protect Maize from Residual Pesticide Injury Caused by PPO-inhibitor Fomesafen”. Those comments are all valuable and very helpful for revising and improving our paper, as well as the important guiding significance to our researches. We have studied the comments carefully and have made correction which we hope meet with approval. The revised part has been marked in red in the revision. The main corrections in the paper and the responds to your comments are as following.

Best regards,

Fei Ye

Response to Reviewer Comments:

I would like to see more precise data about the parameters in the molecular docking study and some signals and values of signals given by the stretching of the ether group.

Response: This question has been modified in the revised manuscript.

This explanation has been added to the revised article as follows:

The parameters are used in the molecular docking study with the default docking setting (Top Hits: 10; Random Conformations: 10; Dynamics Steps: 1000; Forcefield: CHARMm; Ligand Partial Charge Method: Momany-Rone; Final Minimization: Full Potential). (Page 9: 363-365)

The detailed values of signals given by the stretching of the ether group of all compounds have been added to the revised manuscript. (line 117-305) Supplementary Material has been improved and re-uploaded in the revised version.